Comparison between hydroxyapatite and polycaprolactone in inducing osteogenic differentiation and augmenting maxillary bone regeneration in rats

Luchman Nur Atmaliya 1
Megat Abdul Wahab Rohaya 1
Zainal Ariffin Shahrul Hisham 2
Nasruddin Nurrul Shaqinah 3
Lau Seng Fong 4
Yazid Farinawati drfarinawati@ukm.edu.my 1
1 Department of Family Oral Health, Faculty of Dentistry, Universiti Kebangsaan Malaysia , Kuala Lumpur , Malaysia
2 Department of Biological Sciences and Biotechnology, Faculty of Science and Technology, Universiti Kebangsaan Malaysia , Bangi , Selangor , Malaysia
3 Department of Craniofacial Diagnostic and Bioscience, Faculty of Dentistry, Universiti Kebangsaan Malaysia , Kuala Lumpur , Malaysia
4 Department of Veterinary Clinical Studies, Faculty of Veterinary Medicine, Universiti Putra Malaysia , Serdang , Selangor , Malaysia
Leppik Liudmila
Electronic publication date: 2022 May 2
Publication date: 2022
Volume: 10
Electronic Location ID: e13356
Received 2021 Oct 25; Accepted 2022 Apr 8
Copyright: ©2022 Luchman et al.
Copyright year: 2022
Copyright holder: Luchman et al.
License: This is an open access article distributed under the terms of the Creative Commons Attribution License, which permits unrestricted use, distribution, reproduction and adaptation in any medium and for any purpose provided that it is properly attributed. For attribution, the original author(s), title, publication source (PeerJ) and either DOI or URL of the article must be cited.
License URL: https://creativecommons.org/licenses/by/4.0/

Keywords: MC3T3-E1, Hydroxyapatite, Polycaprolactone, Rat, Osteogenesis

Funding: Universiti Kebangsaan Malaysia FRGS/1/2015/SG05/UKM/02/2 This work was supported by Universiti Kebangsaan Malaysia through the Fundamental Research Grant Scheme (FRGS/1/2015/SG05/UKM/02/2) from the Ministry of Higher Education Malaysia. The funders had no role in study design, data collection and analysis, decision to publish, or preparation of the manuscript.

==============================
Background

The selection of appropriate scaffold plays an important role in ensuring the success of bone regeneration. The use of scaffolds with different materials and their effect on the osteogenic performance of cells is not well studied and this can affect the selection of suitable scaffolds for transplantation. Hence, this study aimed to investigate the comparative ability of two different synthetic scaffolds, mainly hydroxyapatite (HA) and polycaprolactone (PCL) scaffolds in promoting in vitro and in vivo bone regeneration.

Method

In vitro cell viability, morphology, and alkaline phosphatase (ALP) activity of MC3T3-E1 cells on HA and PCL scaffolds were determined in comparison to the accepted model outlined for two-dimensional systems. An in vivo study involving the transplantation of MC3T3-E1 cells with scaffolds into an artificial bone defect of 4 mm length and 1.5 mm depth in the rat’s left maxilla was conducted. Three-dimensional analysis using micro-computed tomography (micro-CT), hematoxylin and eosin (H&E), and immunohistochemistry analyses evaluation were performed after six weeks of transplantation.

Results

MC3T3-E1 cells on the HA scaffold showed the highest cell viability. The cell viability on both scaffolds decreased after 14 days of culture, which reflects the dominant occurrence of osteoblast differentiation. An early sign of osteoblast differentiation can be detected on the PCL scaffold. However, cells on the HA scaffold showed more prominent results with intense mineralized nodules and significantly (p < 0.05) high levels of ALP activity with prolonged osteoblast induction. Micro-CT and H&E analyses confirmed the in vitro results with bone formation were significantly (p < 0.05) greater in HA scaffold and was supported by IHC analysis which confirmed stronger expression of osteogenic markers ALP and osteocalcin.

Conclusion

Different scaffold materials of HA and PCL might have influenced the bone regeneration ability of MC3T3-E1. Regardless, in vitro and in vivo bone regeneration was better in the HA scaffold which indicates its great potential for application in bone regeneration.

Introduction

Bone tissue has the ability to spontaneously heal through bone deposition and remodeling (Fernandez-Yague et al., 2015). However, in a larger bone defect due to trauma, surgical treatment of tumor and craniofacial defect such as cleft palate, a bone repair can only be done by bone graft (Fishero et al., 2015; Robey et al., 2015). Bone graft in cleft palate repair is important for tooth eruption and orthodontic tooth movement (Wahab et al., 2020). Craniofacial defect repair by surgeons often requires sophisticated treatment strategies and multidisciplinary input with ideal situations using autologous bone. However, this option is limited by a finite supply of available bone, potential donor site morbidity, particular attention to growing patients, prolonged surgeries in ‘hostile defect‘ that may be associate with free flap loss, anesthetics/patient-related risks, and contour deformities (Lee et al., 2013). In the event of autologous bone being impractical or not feasible, the application of tissue engineering can be a promising concept within the craniofacial surgery field utilizing the engineered materials with a combination of cells to improve or replace biological functions.

Tissue regeneration aims to help the body heal naturally by implanting a scaffold to serve as a temporary matrix that would degrade over time while allowing the regeneration of the host tissue at the implant site. Cellular response and osteoblast differentiation can be affected by morphology, size, surface topography, surface chemistry, porosity, interconnected structure, and fibrous pore wall of the scaffold (Tavakol et al., 2012). Therefore, the selection of scaffold plays a crucial role in ensuring the success of bone regeneration. The chosen scaffold must allow the cells to migrate, proliferate, and differentiate into osteoblasts for the correct development of the bone tissue (Bose, Roy & Bandyopadhyay, 2012). Synthetic scaffolds have advantages over natural scaffolds as they can be manufactured under controlled conditions that allow large scale production with uniform size and design as well as exhibiting reproducible physical and chemical properties (Farinawati et al., 2020).

Hydroxyapatite (HA) is known for its excellent biocompatibility due to its similarity in composition to the apatite found in natural bone. In biological systems, HA occurs as the inorganic constituent for normal calcification such as on bone, teeth, fish enameloid, some species of shell, and in pathological calcification such as on dental and urinary calculus or stone (Hench & Thompson, 2010; Kattimani, Kondaka & Lingamaneni, 2016). Natural occurring HA appears to be brown, yellow, or green in coloration while pure or synthetic HA appears in white coloration. HA contains only calcium and phosphate ions, therefore, no adverse local or systemic toxicity has been reported in any study (Kattimani, Kondaka & Lingamaneni, 2016). Biocompatibility, bioactivity, osteoinductivity, and osteoconductivity are good properties of HA that make them extensively being used as a scaffold for bone regeneration. Moreover, the different forms of HA scaffold that are actively being used include granules (Dorozhkin, 2015), paste and cement (Ben-Nissan, 2014), coatings (Eliaz & Metoki, 2017), porous (Al-Naib, 2018), and dense blocks (Megat Wahab et al., 2020). Nonetheless, concerns have been raised regarding the brittleness and limited degradation properties of HA, including the slow degradation rate (Fiume et al., 2021).

Polycaprolactone (PCL) is much preferred in terms of degradation. PCL is a synthetic polymer that can undergo degradation by hydrolysis of ester bonds in physiological conditions. PCL is an aliphatic semi-crystalline polymer with a melting temperature above body temperature. Hence, at physiological temperature, PCL attains a rubbery state resulting in its high toughness and superior mechanical properties (Dwivedi et al., 2020). PCL appears to be non-toxic and tissue compatible which makes it suitable as scaffolds for bone regeneration. Dwivedi et al. (2020) believed that PCL has easy availability, is relatively inexpensive, and can be modified to adjust its chemical and biological properties, physiochemical state, degradability, as well as mechanical strength. PCL exhibits a degradation time of approximately two to three years and it can be degraded by microorganisms or under physiological conditions (Anderson & Shive, 1997). Its degradation time makes it appropriate for the replacement of hard and load-bearing tissues by enhancing stiffness while decreasing the molecular weight and degradation time for soft tissues. However, several reports have shown that PCL is lack of osteoconductive property due to its poor hydrophilic nature (Hajiali, Tajbakhsh & Shojaei, 2018; Torres et al., 2017; Zhao et al., 2015).

There is still a need to investigate the biological performance of HA and PCL scaffolds in terms of bone integration between the implanted scaffold and surrounding host tissues as well as the difference of bone retention in the defect between these scaffolds. In addition, the question arises of whether different types of scaffold material will affect the cellular and osteogenic potential of the transplanted cells. The effects after transplantation such as tissue rejection and bone viability also need to be considered in determining the success of bone regeneration. Therefore, this study assessed the potential of HA and PCL scaffolds in supporting in vitro cell viability, attachment, morphology, and osteoblast differentiation compared to the accepted model outlined for two-dimensional (2D) systems. Preosteoblast MC3T3-E1 cells were used as cell sources and were cultured on scaffolds and 2D culture plates. The comparison during in vivo bone regeneration of HA and PCL scaffolds was also assessed in this study using a rat model with a maxillary bone defect.

Materials & Methods

Scaffolds preparation

HA scaffold was obtained from GranuMaS® (Granulab, Selangor, Malaysia) with a granule size range from 0.2–1.0 mm. Meanwhile, PCL scaffold was obtained from Osteopore™ (Singapore) with a size range from 3 mm × 1.5 mm (diameter by height). Both HA and PCL scaffolds were sterilized using 75% (v/v) ethanol for 30 min, washed three times in sterile phosphate buffer saline (PBS) (Gibco, Thermo Fisher Scientific, USA), and exposed to 15 min of ultraviolet radiation for each side of the scaffolds. Sterilized scaffolds were immersed in α-Minimum Essential Medium (α-MEM) (Gibco, Thermo Fisher Scientific, USA) overnight prior to the cell culture.

Characterization of scaffolds

HA and PCL scaffolds were sputter-coated with gold in order to obtain sufficient conductivity on the surface and to avoid surface charging during the viewing process. The morphology of the coated scaffolds was viewed under a field emission scanning electron microscope (FESEM, Supra 55VP, Zeiss) and elemental analysis was conducted using energy dispersive X-ray (EDX).

Cell culture

Mouse MC3T3-E1 subclone 14 preosteoblast cells (ATCC No: CRL-2594™) were cultured in a complete medium consisting of α-MEM supplemented with 10% (v/v) fetal bovine serum (Gibco, USA), 1 mM sodium pyruvate (Sigma, USA) and 1% (v/v) penicillin-streptomycin (Gibco, USA). The 2D culture of MC3T3-E1 was conducted as previously done by Yazid et al. (2019). The scaffolds and 96-well plates were seeded with 50,000 MC3T3-E1 cells suspended in a complete medium and incubated for overnight to permit cell attachment. MC3T3-E1 cells-seeded scaffolds were then transferred to new 96-well plates to prevent a false positive result from cells attached at the bottom of wells. All cultured cells were maintained in a humidified atmosphere of 5% (v/v) CO2 at 37 °C and the medium was changed every three days.

For osteoblast differentiation, MC3T3-E1 cells on HA scaffolds, PCL scaffolds, and 2D culture plates were cultured in osteogenic media (α-MEM supplemented with 10% (v/v) fetal bovine serum, 1% (v/v) penicillin-streptomycin, 50 µg/mL ascorbic acid (Sigma, USA) and 10 mM β-glycerophosphate (Sigma, USA)). Meanwhile, MC3T3-E1 cells on scaffolds and 2D culture plate in a complete medium without differentiation factors were used as a negative control for osteoblast differentiation. The differentiation and complete medium were changed every three days.

In vitro analysis

MTT assay for cell viability

The viability of MC3T3-E1 cells on scaffolds and 2D culture plates at 80–90% confluency was evaluated on days 0, 7, 14, and 21 of culture by using the 3-(4,5-dimethylthiazol-2-yl)-2,5-diphenyltetrazolium bromide (MTT) substrate (Sigma, USA) which was reduced to formazan that accumulated in the cytoplasm of viable cells. Briefly, MTT solution (5 mg/mL) and complete medium at a ratio of 1:9 were added to each well containing MC3T3-E1 cells. Cells with MTT solution were incubated at 37 °C in a humidified atmosphere for 4 h. After incubation, the MTT solution was removed and a glycine buffer solution containing dimethyl sulfoxide (Sigma, USA) was added to dissolve formazan salts produced by the enzymatic reaction. After 10 min of agitation, the supernatants were collected and transferred into a new 96-well plate. Then, the absorbance at 570 nm was measured with an ELISA microplate reader (Varioskan Flash Model 680, Thermo Fisher, USA). A viable cell number was obtained through a standard calibration curve determined by correlating a known cell number with the optical density of the solution. For the standard calibration curve, an increasing number of cells from 100, 500, 1,000, 5,000, 10,000. 50,000, 100,000, 500,000 and 1,000,000 of MC3T3-E1 cells were directly cultured and assessed. MTT optical density was normalized to the number of cells on scaffolds and 2D culture plates. MTT assay was performed for five biological replicates and the technical tests were run in triplicates.

Cell attachment and morphology

MC3T3-E1 cell attachment and morphology on HA and PCL scaffolds were examined using FESEM. EDX spectroscopy was conducted to analyze the elemental composition on HA and PCL scaffolds after 21 days of osteoblast differentiation. At osteoblast differentiation culture intervals of 0, 7, 14, and 21 days, cell-seeded scaffolds were fixed overnight in 2.5% (v/v) glutaraldehyde (Polysciences, Inc., Warrington, PA, USA) with PBS and stored at 4 °C. The fixed samples were then washed with PBS three times and subjected to sequential dehydration for 10 min in a graded ethanol series (30% (v/v), 50% (v/v), 70% (v/v), 80% (v/v), 90% (v/v), and 100% (v/v)). Samples of HA scaffold were dried using a critical point drying while samples of PCL scaffold were allowed to dry in air for 24 h at room temperature. Both samples were sputter coated (Quorum, Q150RS) for 30 s with gold and observed under FESEM, at 3–15 kV accelerating voltage.

Alkaline phosphatase specific activity and total protein content

The ALP specific activity was measured by Sensolyte® pNPP alkaline phosphatase assay kit (AnaSpec, USA) according to the manufacturer’s protocol. Briefly, the cells were homogenized in the lysis buffer provided in the kit. Lysate produced was centrifuged for 10 min at 2,500 g at 4 °C. The supernatant was collected and incubated with p-nitrophenyl phosphate (pNPP) at 37 °C for 30 min. Stop solution was added after 30 min of incubation and absorbance measurement was taken at a wavelength of 405 nm using an ELISA microplate reader. Total protein content was measured by Bradford assay with bovine serum albumin used as a standard (Kruger, 2009). ALP activity results were normalized to the total protein content and were represented as U/mg. All samples were run for five independent experiments and repeated three times.

In vivo analysis

Cells and scaffolds preparation

MC3T3-E1 cells were cultured in vitro on HA and PCL scaffolds for 14 days before transplantation. Cells on scaffolds were cultured in a complete media supplemented with osteoblast differentiation factors. The medium was changed every three days. After 14 days of in vitro culture, cells-seeded scaffolds were transplanted into rats with a surgically made maxillary bone defect.

Animals

A total of 24 mature female Sprague Dawley rats (age: 6–8 weeks, body weight: 200–300 g) were used in this study. The housing, care, and experimental protocol were approved by the Universiti Kebangsaan Malaysia Animal Ethical Committee with the approval number FD/2018/ROHAYA/26-SEPT.-2018-JUNE-2019. The animal study was reported according to the ARRIVE guidelines concerning the relevant items. The rats were obtained from the Laboratory Animal Resource Unit, Faculty of Medicine, Universiti Kebangsaan Malaysia (UKM). Prior to the transplantation’s surgery, the rats’ health was monitored for a week. All of the rats were kept in pairs per cage at the animal house of the Faculty of Health Science, UKM, with 12 h light-dark cycle at 21–25 °C and were fed with food pellets. Water was supplied on an ad libitum basis. The activity of rats was observed once daily throughout the study.

Surgery and transplantation

To create the bone defect on the rat’s left maxilla, each rat was first anesthetized with an intravenous injection of 80 mg/kg ketamine (TROY Laboratories PTY Limited, Glendenning, Australia) combined with 7.5 mg/kg xylazine (Indian Immunological Limited, Telangana, India) and 12 mg/kg tramadol (Y.S.P, Kuala Lumpur, Malaysia). A buccal sulcular incision was made to expose the maxillary bone. A bone defect with four mm length and 1.5 mm depth was created in the anterior part of the left maxilla using a trephine bur under constant irrigation. Constant irrigation with cooled sterile PBS was performed to prevent overheating of the bone. The rats were randomly divided into four groups each containing six animals (n = 6). The four groups were as follows: implantation of HA scaffolds (group 1); implantation of PCL scaffolds (group 2); transplantation of MC3T3-E1-HA (group 3); and transplantation of MC3T3-E1- PCL (group 4).

After the transplantation, the mucosal flaps were closed with a simple interrupted suture pattern using 4-0 non-absorbable black silk suture. Postoperatively, each rat received a subcutaneous injection of 2 mg/kg dexamethasone (Duopharma, Malaysia) and 20 mg/kg amoxicillin (Duopharma, Malaysia) for a week to prevent tissue rejection and perioperative infection. The rats were maintained on a soft high-glucose diet for a week. A regular diet was resumed one week postoperatively. Rats were monitored daily by visual observation for signs of infection, inflammation, lack of food and water intake, as well as lethargy.

Euthanasia and maxilla sample collection

All the surviving animals (n = 24) were sacrificed six weeks after the transplantation surgery. Prior to euthanasia, the rats received an intravenous injection of anesthesia (80 mg/kg ketamine, 7.5 mg/kg xylazine, and 12 mg/kg tramadol). Approximately 10 min after the anesthesia induction, rats were then sacrificed via drug overdosing using a commercial euthanasia solution of 390 mg sodium pentobarbital (Vetoquinol SA, France) and 50 mg/ml sodium phenytoin (Duopharma, Malaysia) that were administered intravenously at the lateral vein. Maxilla with intact surrounding tissues from all the rats was dissected and immediately placed in 10% (v/v) neutral buffered formalin (R&M Chemical, UK) for 24 h before being rinsed with PBS. The samples were maintained in a buffer solution consisting of PBS with penicillin and streptomycin at 4 °C until further use.

Micro-computed tomographic analysis

The maxillary section of the rats was scanned using in vivo high-resolution micro-computed tomography (Skyscan 1176; Skyscan, Belgium). The micro-computed tomography (micro-CT) analysis was performed as previously described by Kim et al. (2013). Specifically, the micro-CT projection images were acquired at a source voltage of 70 kV, and a current of 142 µA using a 1 mm aluminium filter with a resolution of 12.32 µm pixels. Scanning was performed by a rotation angle of 360° around the vertical axis, camera exposure time of 580 ms, and a rotation step of 0.7°. Each specimen was scanned for a total of 40 min. The micro-CT images were then reconstructed in the CTAnalyser Skyscan software using approximately 400 scan slices per sample with an image pixel size of 17.56 m. The 2-dimensional projections acquired were elaborated to generate the region of interest (ROI). A 3-dimensional ROI of the bone defect was obtained by manually tracing the margin of each bone defect through the software. The ROI of each treatment group (n = 6) was analyzed for percentages of new bone volume (%), bone surface (mm2), and bone surface density (mm−1). The results were assessed by a trained examiner, who was blinded to the experimental groups (S.F.L.).

Hematoxylin and eosin analysis

Tissue samples were decalcified in 10% (v/w) buffered ethylene diamine tetraacetic acid (Sigma, USA) (pH 7.4) for 5–6 weeks then dehydrated with a graded sequence of increasing ethanol concentrations (from 70% to 100%) and embedded in paraffin. Serial 5 µm-thick sections were generated using a microtome and stained with hematoxylin and eosin (H&E) following the standard protocols. The stained section was observed under a light microscope (BX51; Olympus, Tokyo, Japan) and a digital image was obtained using CellB software. The results of new bone formation within the defect area were taken under 40x and 100x magnifications.

Histological level of new bone formation was assessed using a seven-point scale outline by Salkeld et al. (2001): 0 = no incorporation and no new bone formation, 1 = some incorporation and a small amount of new bone, 2 = some incorporation and a moderate amount of new bone formation, 3 = some corporation with new bone formation continuous with host bone and early remodeling changes in new bone, 4 = good graft incorporation and ample new bone, 5 = good graft incorporation of graft and new bone with host and ample new bone, 6 = excellent incorporation and advanced remodeling of new bone with graft and host. The analysis was validated by an experienced pathologist blinded to the study groups (N.S.N.).

Immunohistochemistry

After deparaffinization, 5 µm-thick tissue sections were subjected to immunohistochemistry (IHC) staining for osteogenesis makers of ALP, and osteocalcin (OCN). Rabbit polyclonal antibody against ALP (1:100, ab65834, Abcam, Cambridge, UK), mouse monoclonal antibody against OCN (1:200, MAB1419, R&D Systems, USA), and Mouse and Rabbit Specific HRP/DAB IHC Detection Kit-Micro-Polymer (ab236466, Abcam, Cambridge, UK) were employed for the study. These primary antibodies were diluted in a solution containing 3% (w/v) BSA in PBS. The specimens were incubated with sodium citrate buffer (Sigma, USA) (pH 6.0) and heated in a microwave oven at the lowest temperature for 10 min. Following a 10-minute incubation with 3% hydrogen peroxide solution, the slides were blocked with Protein Block for10 min to block nonspecific antigens. The specimens were then incubated overnight with a primary antibody at 4 °C. After washing the specimens with PBS buffer, the secondary antibody (goat anti-rabbit HRP-conjugate) incubation steps were conducted following the kit instructions. For visualization, the antigens were detected with DAB chromogen supplied by the kit. All sections were counterstained with Mayer’s hematoxylin, and the slides were dried, then, cover-slipped with DPX mounting medium (VWR Chemicals, France). The results were observed and documented using an Olympus BX51 microscope at 100 × magnification.

The immunohistochemical evaluation of the ALP and OCN expression was performed semi-quantitatively using an immunoreactive score (IRS). The IRS was assessed according to a range of 0–12 as a product of multiplication between the percentage of positive cells score (0–4) and staining intensity score (0–3) as done previously by Koerdt et al. (2013) and Ulu et al. (2018) (Table S1).

Statistical analysis

Multiple comparisons for in vitro cell viability and osteoblast differentiation potential of scaffolds as well as in vivo bone morphometric, histology grading, and IRS analyses were evaluated using Bonferroni-corrected by one-way analysis of variance (ANOVA). Values of p less than 0.05 were considered to be statistically significant. The data were expressed as the mean ± standard deviation of five independent experiments for the in vitro study while a repeated experiment using six rats were used as in vivo biological replicates. Statistical analyses were performed using the SPSS 21.0 software (SPSS Inc., Chicago, IL, USA).

Ethical approval

This research was carried out according to the ethical and legal requirements of the Universiti Kebangsaan Malaysia Animal Ethical Committee (UKMAEC). This permission allowed us to use rats as experimental animals while abiding by the legal and ethical guidelines. Experiments utilized rats were performed humanely throughout this research. The euthanasia method was performed following the guideline from the American Veterinary Medical Association (Underwood & Anthony, 2020). All described experimental protocols involving rats were designed and performed according to the animal ethical guidelines approved by the UKMAEC with approval reference number FD/2018/ROHAYA/26-SEPT./944-SEPT.-2018-JUNE-2019.

Results

The observations and characterizations of HA and PCL scaffolds

FESEM micrographs of HA and PCL scaffolds are presented in Fig. 1A. The HA scaffolds are composed of irregular granules with diameters between 0.2 to 1.0 mm. HA granules showed the morphology of spherical agglomerates with very limited contact areas among granules. The presence of micro-sized grains with a fully interconnected macroporosity could be observed at a higher magnification. Meanwhile, the PCL scaffolds had a typical honeycomb structure and a relatively smooth surface with interconnected equilateral triangles of regular porous morphology. The pore size of the PCL scaffold was measured, and the average pore size was 500 µm. At a higher magnification, the surface of PCL showed the presence of a small groove that may increase scaffold surface area.

Figure 1 Characterization of HA and PCL scaffolds.

(A) FESEM morphology of HA and PCL scaffolds at 23x, 100x, and 1000x magnifications. (B) EDX analysis of HA and PCL scaffolds.

Figure 1B shows the EDX analysis for the HA and PCL scaffolds. The constituents are identified by the peaks concerning their energy levels. HA scaffolds have higher peaks of calcium (Ca) and phosphate (P) than PCL scaffolds. In addition, other essential elements such as carbon (C) and oxygen (O) can also be observed on both scaffolds.

In vitro analysis

Cell viability

MC3T3-E1 cells viability on HA and PCL scaffolds as well as 2D culture plates were measured by the increased number of viable cells throughout the culture period of 21 days. Cells grown in all culture conditions showed a continuous increase with culture time, reflecting good cell viability (Fig. 2). The number of viable MC3T3-E1 cells on scaffolds was markedly higher than the control 2D culture plate, significantly during the initial day of culture (days 0; after a 24-hour attachment period) between MC3T3-E1-HA and MC3T3-E1-2D (p = 0.034). Meanwhile, MC3T3-E1-HA (p = 0.000) and MC3T3-E1-PCL (p = 0.0004) showed a significantly higher number of cell viable on day 7 compared to MC3T3-E1-2D (Table S2). Interestingly, MC3T3-E1 cells grown in HA and PCL scaffolds continued to grow for up to 14 days but this growth decreased at 21 days. MC3T3-E1 cells showed a significantly increased number of viable cells on HA scaffold compared to PCL scaffold on days 7 (p = 0.000) and 14 (p = 0.002) (Fig. 2 and Table S2).

Figure 2 Viability of MC3T3-E1 cells on HA scaffolds, PCL scaffolds, and control 2D culture plates.

The viability of MC3T3-E1 cells on 2D culture plates showed an ongoing increase with culture time, whereas the viability of MC3T3-E1 cells cultured on HA and PCL scaffolds was increased up to 14 days and reduced at 21 days of culture. Values were plotted as a mean number of viable cells ± standard deviation (n = 5).

Figure 3 Osteoblast differentiation potential of MC3T3-E1 cells on HA and PCL scaffolds for 21 days culture period.

(A) Field emission scanning electron microscope image. Arrows indicate mineralized nodules following osteoblast differentiation. Scale bar: 1 µm and 2 µm for HA scaffold, 2 µm for PCL scaffold. (B) EDX analysis for the detection of mineralization. Comparison of EDX elemental analysis on both scaffolds showed a higher level of calcium and phosphorus after 21 days of osteoblast induction which indicates a higher quantity of minerals. (C) ALP specific activity of MC3T3-E1 cells culture on both scaffolds and control 2D culture plates. Asterisks (*) indicate significant differences after a Bonferroni correction for n = 5, at p < 0.05 between different culture conditions. Data represent mean ± standard deviation shown in units per microgram of total cellular protein.

Cell attachment and morphology

FESEM results indicated that both types of scaffolds allowed the attachment and spreading of the cells while maintaining a normal cellular morphology (Fig. 3A). As it can be seen in Fig. 3A, MC3T3-E1 cells on day 0 (a day after osteoblast induction) were already well attached and spread to the surface of both scaffolds, presenting a round shape configuration in HA scaffolds while a cluster of cells with extended cytoplasm in all directions was observed in the PCL scaffolds. After 7 days of culture, cells on both scaffolds showed a typical morphology presenting a flat configuration with more lamellipodia connecting to the neighboring cells and starting to form a continuous cell layer. Mineralized nodules can be observed as early as day 7 in PCL scaffolds. On day 14 of culture, MC3T3-E1 cells on HA scaffolds start to aggregate and form a monolayer of connected cells while a dense cell layer can be seen covering the surface of PCL scaffolds. At 21 days of culture, a dense cell layer could be observed on both scaffolds with some mineralized nodules appearing in between the cell layer especially on HA scaffolds (yellow arrows). During an early culture day, the density of the attached cells is higher in PCL scaffolds compared to HA scaffolds. However, as days of culture increased to day 21, the FESEM image showed that more cells were attached, and an intense appearance of mineralization nodule was observed over HA scaffolds compared to PCL scaffolds.

EDX elemental analysis

EDX spectroscopy results showed the presence of a higher level of calcium and phosphorus after 21 days of MC3T3-E1 cultured on HA and PCL scaffolds (Fig. 3B). HA scaffolds had a higher ratio of calcium/phosphorus (Ca/P) level from 2.29 to 2.4. Meanwhile, PCL scaffolds showed an increase in the Ca/P ratio from 2.22 to 2.36. Both scaffolds showed a slightly higher Ca/P ratio than the theoretical pure hydroxyapatite which is 2.15 (Venkatasubbu et al., 2011). Oxygen and carbon peaks present on the EDX indicate by-products excreted during the extracellular matrix production of MC3T3-E1 cells on the scaffold. The concentration of oxygen and carbon on both scaffolds were observed to be lower than calcium and phosphorus throughout 21 days of osteoblast differentiation.

ALP specific activity

The ALP specific activity was evaluated at days 0, 7, 14, and 21 after MC3T3-E1was cultured in an osteogenic medium (Fig. 3C). A trend of ALP specific activity was increased up to 21 days of osteoblast differentiation in all culture conditions. There were no statistically significant differences (p > 0.05) between MC3T3-E1 cells on scaffolds and 2D culture plates at day 0. The result also showed that the ALP specific activity of MC3T3-E1-PCL (0.19 ± 0.03 U/mg) on day 7 was approximately 1.24 and 3.91 times higher than MC3T3-E1-HA scaffolds (0.15 ± 0.02 U/mg; p = 0.001) and 2D culture plates (0.05 ± 0.03 U/mg; p = 0.000). Interestingly, prolonged osteoblast induction of MC3T3-E1 cells on HA scaffolds resulted in a significantly higher ALP specific activity compared to 2D culture plates especially on days 14 (p = 0.000) and day 21 (p = 0.019) compared to PCL scaffolds.

In vivo analysis

Macroscopic analysis

All animals (n = 24) survived and remained healthy for the entire six weeks of the transplantation period, showing no noticeable sign of toxicity or other adverse effects. Moreover, examined samples present no significant complications such as dehiscence or fistula in the area of the maxilla defect. The maxillary bone defects were healing well without the presence of necrosis or obvious inflammation detected in any fresh maxilla specimen. A stable scaffold fixation with no migration was observed in all rat samples. HA and PCL scaffolds remained after six weeks of transplantation. Some parts of the HA and PCL scaffolds surface were covered by callus (Fig. 4).

Figure 4 Photographs illustrating the surgical transplantation of MC3T3-E1-HA and MC3T3-E1-PCL.

The surgically-made bone defect was created in the anterior part of the rat’s left maxilla. (B) A maxillary bone defect measuring four mm length and 1.5 mm depth was created. (C) Defect received transplantation of MC3T3-E1-HA. (D) Defect received transplantation of MC3T3-E1-PCL. (E) Harvested tissue samples of HA scaffold. (F) Harvested tissue samples of PCL scaffold. Throughout the study period, animals showed a good healing response without adverse tissue reactions. Red circles indicate the presence of scaffolds after six weeks of transplantation.

Micro-computed tomographic analysis

Bone regeneration potential of HA and PCL scaffolds were also investigated using rat’s maxillary bone defect. Surrounding tissues with scaffolds were imaged and analyzed using high resolution micro-computed tomographic (micro-CT), and 2D images were reconstructed (Fig. 5A). Limited to no bony bridge could be observed on empty scaffold groups. As shown in the 2D reconstructed image, treatment of rat maxillary bone defect with MC3T3-E1-HA (group 3) and MC3T3-E1-PCL (group 4) demonstrated bony bridges with an increased amount of filled new bone compared to the control groups. Bridging of the defects with new bone occurred extensively in group 3. Some of the HA granules have consolidated and its radio density increased. Group 4 showed minimal new bone formation and lower radio-density. Defects in group 4 showed a translucent bony bridge at six weeks postoperatively.

Figure 5 Micro-CT analysis showed maxillary bone defect area with the transplantation of cells-seeded scaffolds displaying new bone and defect area after six weeks.

(A) 2D reconstructed image of rat’s maxillary bone treated with scaffolds. No bony bridge formation could be observed in the empty HA and PCL control groups. Bridging of the defects with new bone occurred extensively with increase radio-density on the MC3T3-E1-HA group. No residual of the PCL scaffold can be observed during scanning. Moderate new bone formation and lower radio density with translucent bony bridge could be observed on the MC3T3-E1-PCL group. (B) Graphs display bone morphometric analysis in the form of a percentage of new bone volume, bone surface area, and bone surface density. Data were expressed as mean ± standard deviation. Asterisks (*) indicate significant differences after a Bonferroni correction for n = 6, at p < 0.05.

As shown in Fig. 5B, bone morphometric analysis of the micro-CT images was used to quantify the percentage of the total new bone volume, new bone surface area, and the surface density of newly formed bone. Rats on treatment group 3 (42.74% ± 9.45%) showed a significantly increased new bone volume compared to group 4 (5.43% ± 1.82%; p = 0.002), HA scaffolds control group (12.8% ± 7.08%; p = 0.012) and PCL scaffolds control group (2.22% ± 0.36%; p = 0.001). There is no significant difference (p > 0.05) observed in the increment of bone surface area between treatment groups. Meanwhile, the new bone surface density level of group 3 (7.91 ± 1.07 mm−1) was increased significantly compared to group 4 (1.69 ± 0.29 mm−1; p = 0.001), HA scaffolds control group (3.45 ± 1.43 mm−1; p = 0.014) and PCL scaffolds control group (0.9 ± 0.1 mm−1; p = 0.000).

Microscopic analysis

New bone formation and biocompatibility of the chosen scaffolds in rat’s maxillary bone defect model were evaluated histologically using H&E staining. Defects treated with HA and PCL scaffolds showed cells and tissues infiltration with new bone formation through six weeks of transplantation period (Fig. 6). The newly formed bone in the defect area of MC3T3-E1-HA (groups 3) and MC3T3-E1-PCL (group 4) was higher compared to the empty scaffold control groups. The connective tissue within the bone bridge in groups 3 and 4 was less prominent than in empty scaffolds control groups.

Figure 6 Histological analysis on rat’s maxillary bone defect after six weeks of transplantation.

Histological images from each treatment group at low magnification and high magnification. The arrow symbol indicates newly formed bone. The new blood vessel is marked by an arrowhead. Scale bar 200 µm and 100 µm. HA, hydroxyapatite; PCL, polycaprolactone.

Empty HA scaffold groups showed the defect was surrounded by fibrous connective tissue and the new bone formation was growing from the edge of the cavity towards the center. A moderate amount of bone islands with numerous osteoblast cells as well as a small number of osteocytes in the irregular lacuna could be observed at the edge of the defect. Group 3 showed considerably enhanced new bone formation compared to the empty HA control group. Effective scaffolding property of HA in osteoconduction could be seen with new bone ingrowth that was well developed throughout the pore channels of the HA scaffolds. This can be observed with connective tissues migrated within HA scaffolds. Significant deposition of osteoblast at the marginalized parts of the new bone surrounding the periphery of HA scaffolds could also be observed extensively in group 3 than the empty HA control group.

The empty PCL group showed small new bone formation in the middle of the defect areas with the appearance of extensive fibrous connective tissue ingrowth. Group 4 showed a presence of new bone that was extended into the scaffolds from the edge of the defect towards the center. A moderate amount of bone islands could be observed peripherally. This confirmed that bone formation started at the periphery of the PCL scaffolds. However, intense fibrous and connective tissues were still present in the defect region treated with PCL scaffolds compared to HA scaffolds. Granulation tissue and neovascularization were visible at the defect areas treated with both scaffolds. Both scaffolds demonstrated good biocompatibility with no significant inflammatory reaction.

Histological grading for new bone formation in the defect area treated with MC3T3T-E1-HA and MC3T3T-E1-PCL were higher compared to the empty scaffold control groups. In comparison to their respective controls, the MC3T3-E1-PCL scaffold showed the highest new bone formation with a 1.64-fold increase while only a 1.25-fold increase was observed on MC3T3-E1-HA scaffold. Defects treated with HA scaffolds showed the highest histological grading of new bone formation compared to PCL scaffolds. However, the mean difference between scaffold treatment groups was not statistically significant (p > 0.05) (Table 1).

Immunohistochemistry evaluation

The ability of HA and PCL scaffolds to enhance in vivo bone regeneration was further evaluated by immunohistochemistry analysis. IHC staining revealed that all treatments group treated with HA and PCL scaffolds showed positive expression for ALP and OCN (Fig. 7A).

For both scaffold materials, ALP immunoreactive areas appeared mainly in osteoblast at the interface between newly formed bone and scaffolds. Fibroblasts of the connective tissue were mainly unstained or weakly stained. However, fibroblasts close to the scaffolds also showed weak to moderate immunoreactivity. The expression of ALP was more intensive in treatment groups with HA scaffolds compared to PCL scaffolds prominently at the edge of a new bone, scaffolds interface, and in the extracellular matrix.

OCN immunoreactive area was localized in the osteoblasts, osteocyte, and unmineralized fibrous matrix. Fibrous connective tissue was strong to moderately immunostained. Stronger expression of OCN was observed primarily in the peripheral parts of scaffolds and interfaces between the scaffold and newly formed bone. The immunoreactivity pattern of OCN was similar to ALP, except for intense staining at the interfaces between scaffolds. A newly formed bone and within the connective tissue could be noticed significantly in treatment groups with HA scaffolds rather than PCL scaffolds. Connective tissues inside the HA scaffolds also showed positive expression of OCN.

Table 1 Histological grade of bone formation at maxillary bone defect after six weeks of transplantation. Data expressed as mean difference ± standard deviation (n = 6).

No significant (p > 0.05) difference for MC3T3-E1-HA and MC3T3-E1-PCL compared to empty HA and PCL control groups.

Treatment groups	New bone formation (0 to 6)	
Empty HA scaffold	2.00 ± 0.68	
Empty PCL scaffold	1.83 ± 0.60	
MC3T3-E1-HA scaffold	2.50 ± 0.43	
MC3T3-E1-PCL scaffold	3.00 ± 0.52	

Figure 7 Immunohistochemistry analysis on rat’s maxillary bone defect after six weeks of transplantation.

(A) Immunohistochemical staining for alkaline phosphatase (ALP) and osteocalcin (OCN) in each treatment group. The arrow symbol shows strong positive staining. Scale bar 100 µm. (B) Immunoreactive score (IRS) for ALP and OCN. Asterisks (*) indicate significant differences between different treatment groups after a Bonferroni correction at p < 0.05. Data represent the mean IRS ± standard deviation of the six rats observed in at least two different visual fields.

Mean IRS for ALP increased significantly in the MC3T3-E1-HA and MC3T3-E1-PCL compared to the empty HA scaffold (p = 0.001 and p = 0.002, respectively) (Fig. 7B). Moderate IRS grade for ALP was demonstrated prominently on MC3T3-E1-HA followed by MC3T3-E1-PCL, PCL, and HA. Meanwhile, MC3T3-E1-HA and MC3T3-E1-PCL showed a significantly higher mean IRS for OCN compared to empty PCL scaffolds groups (p = 0.010 and p = 0.026, respectively) (Fig. 7B). Moderate IRS grade for OCN was demonstrated prominently on MC3T3-E1-HA followed by MC3T3-E1-PCL, HA, and PCL.

Discussion

In this study, MC3T3-E1 cell viability on HA scaffold, PCL scaffold, and 2D culture plate was measured and compared to confirm the ability of chosen scaffolds to support the growth of cells during 3D in vitro culture. Through MTT assay, we identified that the number of viable MC3T3-E1 cells on scaffolds was significantly higher than the control 2D culture plate even during the initial day of culture, indicating that the 3D structure of scaffolds may provide an optimum growth environment for cells by facilitating more space for nutrient and metabolic waste exchange (Hoveizi et al., 2014; Lim, Sun & Sultana, 2015; Mehendale et al., 2017). Moreover, the viability of MC3T3-E1 cells on the HA scaffold was significantly increased with a peak number of viable cells observed on day 14. The cellular response and cell’s behavior can be influenced by the characteristic of the material surface (Seebach et al., 2010; Shamsuddin et al., 2017). The HA scaffold used in this study exhibits a rough surface that may enhance cell growth. A previous study by Ling et al. (2015) demonstrated that cells seeded on the HA-composite scaffold had higher cell proliferation compared to the β-tricalcium phosphate-composite scaffold due to the rough texture present on the surface of HA. This report is consistent with our finding that the HA scaffold promotes higher cell growth during the initial period. However, it was significantly decreased with a prolonged culture which reflects the dominant occurrence of osteogenic differentiation. This pattern has also been observed in cell viability on the PCL scaffold. The obvious reduction in cell viability on HA and PCL scaffolds compared to control 2D culture plate may be due to the transition of the cells from a proliferative phase to the differentiation phase induced by direct interactions with the scaffolds (Weissenböck et al., 2006). This finding suggests that HA and PCL scaffolds are cytocompatible as they support and enhance MC3T3-E1 cell viability.

MC3T3-E1 cells differentiation toward osteoblast was observed through FESEM image, EDX analysis, and ALP specific activity. FESEM image showed intense attachment, well-spread morphology and extensive growth of osteoblast differentiated MC3T3-E1 cells on both scaffolds. These results coincide with studies by of Seebach et al. (2010) and Jamal et al. (2018) who also found strong attachment, growth, and proliferation of mesenchymal stem cells on HA and PCL scaffolds. Although the HA has the potential to naturally induce osteoblast differentiation, MC3T3-E1 cells on the PCL scaffold showed signs of osteoblast differentiation as early as day 7 of induction. On day 7 of osteoblast differentiation, mineralization nodules were detected on the PCL scaffold while it was not present on the HA scaffold. When cells come into proximity to one another, presumably, they are stimulated to differentiate and mineralization begins once multilayers of cells formed (Bellows et al., 1986; Jamal et al., 2018). A study by Bellows et al. (1986) demonstrated that the ability of cells to form multilayers seems to be the basis of mineralization due to the incapability of single-cell layers to produce a mineralized matrix. In this study, MC3T3-E1 cells on PCL scaffolds at day 7 became close to each other as more cells can be observed attaching to the scaffold compared to HA scaffolds. These results are in agreement with Jamal et al. (2018) which demonstrated early osteoblast differentiation of dental pulp stem cells on PCL scaffolds was influenced by intense cell-to-cell contact. However, mineralized nodules can be observed more prominently on HA scaffolds with prolonged osteoblast differentiation. These results are likely to be related to the rough texture present on the surface of HA that coherently supports extensive MC3T3-E1 cell growth. An increased number of MC3T3-E1 cells may permit more cell-to-cell contact that directly induced osteoblast differentiation and mineralization. This finding raises the possibility that the surface characteristic of the scaffold could provide a suitable microenvironment for cell attachment, cellular interaction, and osteoblast differentiation.

These results were further supported by EDX analysis which allowed assessment and quantification of the presence of different bone types based on elemental analysis of calcium, phosphorus, and nitrogen (Prati et al., 2020). Therefore, EDX spectroscopy was utilized to detect the amount of elemental calcium and phosphorus produced by cells grown on HA and PCL scaffolds. EDX results demonstrated that MC3T3-E1 cells were able to attach and grow on HA and PCL scaffolds and form mineralized nodules or tissues consisting of calcium and phosphorus deposits. HA scaffold showed the higher ratio of calcium to phosphorus after 21 days of osteoblast differentiation which indicates a higher quantity of minerals.

In addition to the initial evaluation of MC3T3-E1 cell morphology using FESEM following osteoblast differentiation, an increase in ALP specific activity of these cells on scaffolds is evidence of successful induction of osteogenic differentiation. ALP is one of the generally recognized biochemical markers for bone cell activity and is considered to play a role in bone mineralization (Megat Wahab et al., 2020). Scaffolds for bone regeneration application must support the differentiation of cells to functional bone tissue. Our results showed an overall increase of ALP specific activity throughout 21 days of osteoblast differentiation for all culture conditions. Higher ALP specific activity of MC3T3-E1 cells has been observed on HA and PCL compared to the control 2D culture plate. Both scaffolds were able to support osteoblast differentiation of MC3T3-E1 cells. The results of this study also showed that MC3T3-E1 cells on PCL scaffolds required a shorter time to undergo osteoblast differentiation with a high ALP specific activity detected as early as day 7. However, as the day of osteoblast differentiation increased, ALP specific activity of MC3T3-E1 cells on HA scaffolds showed a more prominent result. A previous study by Deligianni et al. (2000) suggested that there was a delay in the expression of ALP activity on HA scaffolds with rougher surfaces. These results are consistent with the FESEM images that showed mineralized nodules of MC3T3-E1 cells on PCL can be observed as early as day 7 and an increase of mineralized nodules was observed with a prolonged culture on HA scaffolds. An increase in ALP specific activity of MC3T3-E1 cells on HA scaffolds at the end of the osteoblast differentiation period was further enhanced by osteoinductive property present on the scaffolds. Studies by Usui et al. (2010) and Wang et al. (2015) indicate the potential of calcium and phosphate ions released from HA could directly induce and up-regulates osteoblast differentiation which promotes bone formation through calcification. Taken together, these findings suggest the potential of HA and PCL scaffolds in promoting in vitro osteoblast differentiation which can be observed more prominently in HA scaffolds.

Since this study also aims to test different scaffolds material on enhancing in vivo bone regeneration, it is important to consider that our animal model is ideal for scaffold implantation in the maxillary bone defect treatment. No synchronization for rat oestrus cycles was performed in this study, which may have a significant effect on osteogenesis. Even though additional studies in more diverse populations of animal gender should be conducted, recent analyses of variability in male and female’s animal models demonstrate that unstaged females are not more varied than males across diverse traits, from gene expression to hormone levels in multiple species (Becker, Prendergast & Liang, 2016; Dayton et al., 2016; Itoh & Arnold, 2015; Prendergast, Onishi & Zucker, 2014; Smarr et al., 2017). Beery (2018) suggested that estrous variability in a female animal is no greater than intrinsic variability in males. The variability may arise from different sources such as consistent estrous-cycle dependent variability that may be exhibited by female animals, while male animals showed variability on different timescales, or more variability between individuals. No animal death was observed during the transplantation period. Postoperatively, all rats showed immediate recovery as daily activities resumed within 24 h. There were no complications such as dehiscence or fistula in the area of transplantation. A stable scaffold fixation with no migration was observed in all rat samples. HA and PCL scaffolds remained after six weeks of transplantation. This indicates that both scaffolds have slow biodegradation under physiologic conditions with high mechanical resistance. A similar finding was also reported by Wongsupa et al. (2017) which suggests that slow biodegradation of PCL-biphasic calcium phosphate scaffold could have advantages for highly loaded areas that required long-term critical functional support.

In the presented study, in vivo bone regeneration of transplanted MC3T3-E1 cells on HA and PCL scaffolds were evaluated by micro-CT and histology analyses. The reconstructed micro-CT image revealed a radio-dense appearance, suggesting bone ingrowth within HA and PCL scaffolds that filled the defect area. These results corroborated the ideas of Crovace et al. (2020) who showed that new bone formation could be observed by radio-dense aspect from X-ray images. Meanwhile, the bone morphometric analysis revealed that HA scaffolds significantly enhanced new bone volume and surface density, although no significant differences was found on bone surface area compared to PCL scaffolds. Our results are similar to a previous study by Jang et al. (2017), which found bone formation was indicated by a higher bone volume present in granular and porous HA scaffolds. Low bone surface area occurs when trabecular thickness, bone surface density to tissue volume are at a high level (Kim et al., 2004). Micro-CT can be over-estimated depending on the characteristics of bone substitutes or scaffolds. Therefore, there are chances that an increase in bone formation is due to the inherent radio-opacity contained in HA scaffolds (Jang et al., 2017). Meanwhile, the low value of bone volume, surface area, and surface density on PCL scaffold is due to the translucent property of PCL. This led us to use a qualitative histological approach to further evaluate the new bone formation.

The results from the histological observation showed that HA and PCL scaffolds are capable of enhancing new bone formation with intensive vascularity at the defect area, indicating the bone vitality and beginning of new bone formation. Both scaffolds revealed good biocompatibility with no adverse inflammatory side effects. Furthermore, the defect area with transplanted HA scaffold showed extended bone formation not only on its surface but also in the pores of the scaffold with the infiltration of connective tissues present within HA particles. This showed that the HA scaffolds used in this study is a porous scaffold that allows cell migration and proliferation. The presence of connective tissue inside the HA scaffold may also resulted from the interaction of MC3T3-E1 cells and host bone-forming cells with the osteoconductive property of HA scaffolds. Wypych (2018) referred to osteoconduction as the ability of bone-forming cells in the grafting area to move across a scaffold and slowly replace it with a new bone over time. These results are in agreement with Sulaiman et al. (2013) and Shao et al. (2018) that demonstrated bone formation for ceramic scaffold, started on the surface and proceeded to the center of the pores. Meanwhile, bone tissue formed on PCL scaffold as a result of the extensive proliferation of differentiation of cells from the surface of the PCL fibers that further grow in between several fibers and encapsulate them over time. There is no evidence to suggest that PCL has osteoinductive or osteoconductive properties. Therefore, factors such as enhanced cell-to-cell and cell-to-substrate interaction might contribute to the observed new bone formation on the PCL scaffold (Rumiński et al., 2018).

Nonetheless, H&E staining is insufficient to conclude the types of tissue relevant to osteogenesis. Hence, the identification of new bone formation in scaffolds during in vivo study should be determined using more complex staining methods. However, in this study, new bone formation was not evaluated using Masson-Goldner trichrome, modified Masson-Goldner trichrome, or Movat’s pentachrome staining but by a stronger expression of ALP and OCN markers using immunohistochemistry (IHC) analysis. Other studies did not perform complex staining methods but used IHC staining to demonstrate the changes in osteoblast activity by the expression of ALP and OCN markers as an osteogenic maturation and occurrence of mineralization marker at the bone-connective tissue interface (Abbasi et al., 2020; Christenson, 1997; Jeon et al., 2014; Yasui et al., 2016).

The IHC analysis supported micro-CT and histological observation which revealed the HA scaffold has a greater potential to augment maxillary bone regeneration in rats. The IHC staining showed that new bone tissues formed at the maxilla defect area were ALP and OCN positive. ALP is considered an intermediate marker and is secreted during osteoblast maturation and matrix mineralization (Khanna-Jain et al., 2012). Meanwhile, OCN expression is presented as a late marker of osteogenesis and is consistent with characteristics of mature bone (Freire et al., 2015). Stronger expressions of ALP and OCN were observed in all treatment groups that received HA scaffolds suggest that the osteogenic process was more advanced in the HA scaffold than the PCL scaffold. Nevertheless, OCN showed intense expression compared to ALP which prominently could be observed on the MC3T3-E1-HA group followed by MC3T3-E1-PCL, empty HA, and empty PCL groups. Significant differences were found between these markers, especially on the scaffold with cells groups compared to empty scaffold groups. Similar results have been observed in other studies evaluating the performance of HA and PCL-based scaffolds for in vivo bone models (Abbasi et al., 2020; Johari et al., 2016). Thus, the results from these studies strongly support the stronger expression of OCN compared to ALP due to the enhanced osteogenic behavior of the scaffold. An increase in the cell populations residing in the scaffolds could lead to the intense production of osteogenic markers and subsequently accelerate the differentiation process (Zhou et al., 2020). The results from the present study indicate that HA and PCL scaffolds have the potential to repair rat’s maxillary defects with properties applicable for bone tissue engineering. Moreover, it was concluded that the transplantation of cells and HA scaffolds showed better in vivo bone regeneration potential with enhanced new bone volume, increment of bone surface density, and new bone formation as shown by micro-CT, histology, and IHC analyses.

Conclusions

In conclusion, HA and PCL scaffolds used in this study can support in vitro cell viability, attachment, morphology, and osteoblast differentiation following the accepted model of the two-dimensional system. Both scaffolds demonstrated good bone regeneration properties with no significant inflammatory reaction. However, HA scaffolds showed better new bone formation when transplanted on the maxillary bone defect of rats compared to PCL scaffolds. This confirmed in vitro growth and osteoblast potential of HA scaffolds. Based on the obtained result, it is suggested that HA could be considered as a potential scaffold for clinical use in maxillary bone regeneration.

Supplemental Information

Supplemental Information 1 Immunoreactive score (IRS) assigned for the semi-quantitative immunohistochemical evaluation of the ALP and OCN expression

Click here for additional data file.

Supplemental Information 2 Comparison of a number of viable MC3T3-E1 cells when cultured on HA scaffolds, PCL scaffolds, and 2D culture plates

One-way ANOVA was conducted to compare MC3T3-E1 cell viability when cultured on scaffolds and 2D culture plates in terms of a number of viable cells (1 × 104 cells). Values were mean difference ± standard deviation. *Asterisks indicate significant differences after a Bonferroni correction for n = 5, at p < 0.05. MTT assays for cell viability were carried out in triplicate.

Click here for additional data file.

Supplemental Information 3 Raw data

In vitro cell viability analysis using MTT assay as well as ALP specific activity using ALP assay. In vivo analysis comprising micro-CT and hematoxylin, eosin, and immunoreactive score (IRS) analyses are also shown.

Click here for additional data file.

Supplemental Information 4 ARRIVE Checklist

Click here for additional data file.

The authors wish to thank Dr. Siti Khadijah Shuhaimy Basha for her kind help during the animal surgery and transplantation procedure.

Additional Information and Declarations

Competing Interests

Author Contributions

Animal Ethics

Data Availability

The authors declare there are no competing interests.

Nur Atmaliya Luchman performed the experiments, analyzed the data, prepared figures and/or tables, and approved the final draft.

Rohaya Megat Abdul Wahab, Shahrul Hisham Zainal Ariffin and Farinawati Yazid conceived and designed the experiments, authored or reviewed drafts of the paper, and approved the final draft.

Nurrul Shaqinah Nasruddin analyzed the data, authored or reviewed drafts of the paper, and approved the final draft.

Seng Fong Lau analyzed the data, prepared figures and/or tables, and approved the final draft.

The following information was supplied relating to ethical approvals (i.e., approving body and any reference numbers):

Universiti Kebangsaan Malaysia Animal Ethical Committee (UKMAEC) provided full approval for this research (FD/2018/ROHAYA/26-SEPT.-2018-JUNE-2019)

The following information was supplied regarding data availability:

The raw measurements are available in the Supplementary Files.

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
