# Peer review of "Comparison between hydroxyapatite and polycaprolactone in inducing osteogenic differentiation and augmenting maxillary bone regeneration in rats"

_PeerJ, doi:10.7717/peerj.13356_

## Round 0.1 · original submission · Major Revisions

Please provide detailed answers to the reviewers' comments, please also see the annotated manuscript from Reviewer 3.

Reviewer 1 ·

Basic reporting

In this study, Luchman et al aim to investigate the capacity of two different synthetic scaffolds (HA and PCL scaffolds) in promoting in vitro and in vivo bone regeneration. Based on the data collected, they found HA scaffold works better on maintaining cell viability and promoting bone formation. And the authors concluded that HA and PCL scaffold geometry might have influenced on the bone regeneration capacity of MC3T3-E1 cells. However, HA scaffold shows better capacity in promoting bone generation in both in vitro and in vivo experiments, which indicates its great potential for application in bone regeneration.
This study is well-organized, experimental design is easy to follow, and the results are scientifically sound. The biggest highlight in this study is the application of surgical transplantation of MC3T3-E1-HA and MC3T3-E1-PCL, which gives directly evidence that HA scaffold showed better in vivo bone regeneration potential with enhanced new bone volume, increment of bone surface density, and new bone formation.
If it is applicable, this manuscript can be considered for publication in its current form.

Experimental design

Experimental design is good.

Validity of the findings

Findings are scientifically sound.

·

Basic reporting

The author first needs to decide the overall aims of the study and do the design according to that. The main issue in this manuscript is the lack of clarity of the overall aim. In biomaterials, scaffold geometry is referred to, i.e., scaffold pore size, scaffold curvature, and or design. Here, the authors compared different materials, not only differ geometrically.

Experimental design

The author used PCL and HA, then evaluated the effect of scaffolds geometry and surface roughness on cellular attachment; however, scaffold geometry is not the exact thing your study focused on. The study is missing an essential aspect of defining material in terms of chemical characterization, morphology for PCL pore size, and analysis of materials properties.

In line 295, the author stated, ‘The number of viable MC3T3-E1 cells on scaffolds was markedly higher than the control 2D culture plate, significantly during the initial day of culture (days 0) between MC3T3-E1-HA and MC3T3-E1-2D (p = 0.034).’ the authors need to realize that MTT assay reflects the metabolic activity of cells, not a number.

Validity of the findings

The authors referred to a question that was not fully supported by the study design. The title and aim of the study need to be more specific toward the materials used.
The results need to be supported with more material characterization analysis, including SEM images for scaffolds before cells seeding, AFM data to indicate surface roughness, EDS data for elemental analysis.

Additional comments

The author stated that "All of the surviving animals were sacrificed six weeks"-not clear what was the total surviving number and whether it has affected the n=6 indicated in their in vivo study.
The author stated that "It has been proved that the surface topography of rough-textured capable to enhance the cellular adhesion and production of more mineralized matrix during osteoblast differentiation of cells." -Missing reference.
The author stated that "The obvious reduction in cell viability on HA and PCL scaffolds in comparison with control 2D culture plate may be due to the transition of the cells from a proliferative phase into the differentiation phase induced by direct interactions with the scaffold" -Missing reference.
Masson trichrome staining is critical to identify and define new bone formation

Reviewer 3 ·

Basic reporting

1- A proof reading for English language is suggested.
2- The title should be changed. Please see my first comment on "Validity of the findings" section.
3- Some minor revisions are suggested. Please see the last page of the annotated manuscript which I upload (After line 664).
4- Authors are suggested to use marks like arrows and arrow heads in figure 7. For example in the high magnification PCL image, authors placed NB near to an osteoid structure, and the writing is on connective tissue. Inexperienced readers may misunderstand this image therefore, putting pointing marks could be beneficial.

Experimental design

1- Although some additional techniques might be used in the study, the used methodology is acceptable for answering the research question.

2- Is it possible for authors to use more appropriate staining technics for paraffin sections if you still have unstained sections or blocks available? Staining techniques such as Masson Goldner trichrome staining, Movat’s pentachrome staining, etc. are commonly used staining technics to see tissue transitions and identify all tissue types relevant with the osteogenesis which would allow you to address different tissue types much better. For example, in the last image (high magnification MC3T3-PCL) of figure 7, some of the NB areas are probably consist of cartilage and there’s a fibro-cartilage transition area as well but H&E staining is not providing enough details to fully identify these structures.

3- For my detailed comments and further questions, please see the last page of the annotated manuscript (After line 664).

Validity of the findings

1- The title of the manuscript is indicating the effect of “scaffold geometry”, but I believe the used methodology and obtained results are not sufficient to say anything about the effect of geometry. HA and PCL are entirely different materials and as it is mentioned in the manuscript, they have some different qualities effecting the osteogenesis. If the HA and PCL had the same geometry, could we expect to get similar results between the experimental groups? The geometry of scaffolds indeed proved to have good amount of influence but scientists usually use different geometrical patterns with same type of material like using HA scaffolds in hexagonal geometry vs HA scaffolds in square geometry. Therefore, the title must be changed and all the statements related with scaffold geometry should be discarded.

2- Line 420: Line 420: To evaluate cellular adhesion, other technics should be taken into consideration. If you’re making this statement depending upon your electron microscopy imaging results, please refer this analysis in the sentence. Otherwise, it feels like your MTT results provided this knowledge of enhanced adhesion.

3- Line 444: "and once they form multilayers they start mineralization (Yu et al., 2017)."
In this article Yu et al. mentions double-layered cell transfer method (https://www.nature.com/articles/srep33286 ). There’s no mentioning of that the cells start mineralization when they become multilayered. Please revise this statement.

4- Line 466: Changing “successful osteoblast formation” to “successful induction of osteogenic differentiation.” could be a better choice. There are several cellular steps throughout the osteogenic differentiation starting with mesenchymal stem cells to osteocytes and beginning from the early osteoprogenitors, those cells express ALP gene and produce ALP enzyme. Therefore, saying successful osteoblast formation is a bit assertive as these cells could be of any cell type from an early progenitor to a mature osteoblast.

5-Line 522-523:”The infiltration of connective tissue resulted from the interaction between MC3T3-E1 cells and the osteoconductive property of HA scaffold”
Which results are refering to this statement? Connective tissue normally does not infiltrate into an area. In normal physiological healing, cells do migrate to defect area and they differentiate form a fibrous tissue. Afterwards cells do condensate and form fibro-cartilage tissue parts which would enlarge and form a cartilage later on. Cells keep on differentiating and due to vascularization and ALP activity, osteoid structures occur in the defect site. Please change the statement for this part.

6- For my other comments please see the annotated manuscript that I upload (After line 664).

Additional comments

Authıors had investigated the qualities of HA and PCL scaffolds seeded with MC3T3 cells both in vitro and in vivo. Although the study is well planned, I believe the manuscript could use some improvements and some changes also had to be made starting with the title of the study.

Annotated reviews are not available for download in order to protect the identity of reviewers who chose to remain anonymous.

---

## Round 0.2 · Minor Revisions

Please provide the minor revisions suggested by Reviewer 3 concerning IHC and MTT.

·

Basic reporting

The authors have adequately answered all requests.

Experimental design

The authors have adequately answered all requests.

Validity of the findings

The authors have adequately answered all requests.

Additional comments

The quality and clarity of the paper are much improved and I have no further comments at this time. Very nice job!

Reviewer 3 ·

Basic reporting

1. Authors are suggested to avoid self-citations unless it is absolutely necessary (Ariffin et al., 2017; Abdul Wahab et al., 2020; Wahab et al., 2020; Yazid et al., 2019; Yazid et al., 2018).

Experimental design

1. Which solution is used for primary antibody dilutions in IHC? If you used commercial antibody diluents, please refer the product and company names and if you prepared your own conservative solution please mention it.
2. Addition of IHC to the paper definitely added value to the manuscript. Although addition of anti-OCN and anti-ALP stainings are enough to bring good results into the table, there’s still room for improvements. Is it possible for authors to perform a quantitative study for IHC results by using softwares such as ImageJ or at least making a semi-quantitative evaluation by measuring the power of immunoreactivity in the defect zone (like +=weak, ++=moderate, +++=strong reaction)? I believe after gathering quantitative results, investigating if there’s any significant difference between ALP and OCN reactions between the groups would definitely strengthen the manuscript.

Validity of the findings

Authors had made a statement about cellular adhesion regarding to their MTT results. MTT assay is being utilized for measuring the toxicity of used reagents in vitro, yet the logic of the assay also helps researchers to make measurements about cell viability and proliferation in some cases. When I checked the articles, I see that researchers are using MTT assay to evaluate the effect of used material in cellular proliferation rates and using SEM imaging to prove cellular adhesion.
I do understand that having positive results in MTT assay also would mean that the cells are indeed adhered to the material but you are already utilizing electron microscopy imaging for this purpose and we clearly see that the cells are adhered to the surface of materials. So, I believe you should discuss cellular adhesion depending on your electron microscopy results rather than MTT since it’s not a very appropriate technic to show cellular adhesion.

Additional comments

Dear authors;

Thank you for your detailed response to all reviewer comments. I would like to say that I believe adding IHC after making necessary changes in the manuscript really improved the value of your paper. Although it's not vital, I still belive if you can find some time make quantitative or semi-quantitative measurements for IHC staining would add much more value to your discussion. You can see my other minor concerns in my report.

---

## Round 0.3 · Minor Revisions

Please provide minor editings pointed by reviewer.

Reviewer 3 ·

Basic reporting

1-Lines 308-311: Please place a citation for the used scoring system for IRS, if applicable.
2- "Multiple comparisons for in vitro cell viability and osteoblast differentiation potential of scaffolds as well as in vivo bone morphometric and histology grading analysis were evaluated using Bonferroni-corrected by one-way analysis of variance (ANOVA). "

Please mention the used statistical test for IRS. If the same test had been used, please mention that IRS is also subjected to that test.

Experimental design

The authors have answered all requests.

Validity of the findings

The authors have answered all requests.

Additional comments

Thank you for your efforts and answering all the requests in a positive manner. After these 2 little editings I believe there's no further concerns available for your manuscript.
Best Regards.

---

## Round 0.4 · accepted · Accept

The authors provided all necessary corrections and manuscript is now accepted for publication, congratulations!